# Learning in Games: Robustness of Fast Convergence

Dylan J. Foster[*]    Zhiyuan Li[†]    Thodoris Lykouris[*]    Karthik Sridharan[*]    Éva Tardos[*]

## Abstract

We show that learning algorithms satisfying a *low approximate regret* property experience fast convergence to approximate optimality in a large class of repeated games. Our property, which simply requires that each learner has small regret compared to a $(1 + \epsilon)$-multiplicative approximation to the best action in hindsight, is ubiquitous among learning algorithms; it is satisfied even by the vanilla Hedge forecaster. Our results improve upon recent work of Syrgkanis et al. [28] in a number of ways. We require only that players observe payoffs under other players' realized actions, as opposed to expected payoffs. We further show that convergence occurs with high probability, and show convergence under bandit feedback. Finally, we improve upon the speed of convergence by a factor of $n$, the number of players. Both the scope of settings and the class of algorithms for which our analysis provides fast convergence are considerably broader than in previous work.

Our framework applies to dynamic population games via a low approximate regret property for shifting experts. Here we strengthen the results of Lykouris et al. [19] in two ways: We allow players to select learning algorithms from a larger class, which includes a minor variant of the basic Hedge algorithm, and we increase the maximum churn in players for which approximate optimality is achieved.

In the bandit setting we present a new algorithm which provides a "small loss"-type bound with improved dependence on the number of actions in utility settings, and is both simple and efficient. This result may be of independent interest.

## 1  Introduction

Consider players repeatedly playing a game, all acting independently to minimize their cost or maximize their utility. It is natural in this setting for each player to use a learning algorithm that guarantees small regret to decide on their strategy, as the environment is constantly changing due to each player's choice of strategy. It is well known that such *decentralized no-regret dynamics* are guaranteed to converge to a form of equilibrium for the game. Furthermore, in a large class of games known as *smooth games* [23] they converge to outcomes with approximately optimal social welfare matching the worst-case efficiency loss of Nash equilibria (the *price of anarchy*). In smooth cost minimization games the overall cost is $\lambda/(1 - \mu)$ times the minimum cost, while in smooth mechanisms [29] such as auctions it is $\lambda/\max(1, \mu)$ times the maximum total utility (where $\lambda$ and $\mu$ are parameters of the smoothness condition). Examples of smooth games and mechanisms include routing games and many forms of auction games (see e.g. [23, 29, 24]).

The speed at which the game outcome converges to this approximately optimal welfare is governed by individual players' regret bounds. There are a large number of simple regret minimization algorithms (Hedge/Multiplicative Weights, Mirror Decent, Follow the Regularized Leader; see e.g. [12]) that

---

[*]Cornell University {djfoster,teddlyk,sridharan,eva}@cs.cornell.edu. Work supported in part under NSF grants CDS&E-MSS 1521544, CCF-1563714, ONR grant N00014-08-1-0031, a Google faculty research award, and an NDSEG fellowship.

[†]Tsinghua University, lizhiyuan13@mails.tsinghua.edu.cn. Research performed while author was visiting Cornell University.

guarantee that the average regret goes down as $O(1/\sqrt{T})$ with time $T$, which is tight in adversarial settings.

Taking advantage of the fact that playing a game against opponents who themselves are also using regret minimization is not a truly adversarial setting, a sequence of papers [9, 22, 28] showed that by using specific learning algorithms, the dependence on $T$ of the convergence rate can be improved to $O(1/T)$ ("fast convergence"). Concretely, Syrgkanis et al. [28] show that all algorithms satisfying the so-called RVU property (Regret by Variation in Utilities), which include Optimistic Mirror Descent [22], converge at a $O(1/T)$ rate with a fixed number of players.

One issue with the works of [9, 22, 28] is that they use expected cost as their feedback model for the players. In each round every player receives the expected cost for each of their available actions, in expectation over the current action distributions of *all other players*. This clearly represents more information than is realistically available to players in games — at most each player sees the cost of each of their actions given the actions taken by the other players (*realized feedback*). In fact, even if each player had access to the action distributions of the other players, simply computing this expectation is generally intractable when $n$, the number of players, is large.

We improve the result of [28] on the convergence to approximate optimality in smooth games in a number of different aspects. To achieve this, we relax the quality of approximation from the bound guaranteed by smoothness. Typical smoothness bounds on the price of anarchy in auctions are small constants, such a factor of 1.58 or 2 in item auctions. Increasing the approximation factor by an arbitrarily small constant $\epsilon > 0$ enables the following results:

- We show that learning algorithms obtaining fast convergence are ubiquitous.
- We improve the speed of convergence by a factor of $n$, the number of players.
- For all our results, players only need feedback based on realized outcomes, instead of expected outcomes.
- We show that convergence occurs with high probability in most settings.
- We extend the results to show that it is enough for the players to observe realized *bandit* feedback, only seeing the outcome of the action they play.
- Our results apply to settings where the set of players in the game changes over time [19]. We strengthen previous results by showing that a broader class of algorithms achieve approximate efficiency under significant churn.

We achieve these results using a property we term Low Approximate Regret, which simply states that an online learning algorithm achieves good regret against a multiplicative approximation of the best action in hindsight. This property is satisfied by many known algorithms including even the vanilla Hedge algorithm, as well as Optimistic Hedge [21, 28] (via a new analysis). The crux of our analysis technique is the simple observation that for many types of data-dependent regret bounds we can fold part of the regret bound into the comparator term, allowing us to explore the trade-off between additive and multiplicative approximation.

In Section 3, we show that Low Approximate Regret implies fast convergence to the social welfare guaranteed by the price of anarchy via the smoothness property. This convergence only requires feedback from the realized actions played by other players, not their action distribution or the expectation over their actions. We further show that this convergence occurs with high probability in most settings. For games with a large number of players we also improve the speed of convergence. [28] shows that players using Optimistic Hedge in a repeated game with $n$ players converge to the approximately optimal outcome guaranteed by smoothness at a rate of $O(n^2/T)$. They also offer an analysis guaranteeing convergence of $O(n/T)$, at the expense of a constant factor decrease in the quality of approximation (e.g., a factor of 4 in atomic congestion games with affine congestion). We achieve the convergence bound of $O(n/T)$ with only an arbitrarily small loss in the approximation.

Algorithms that satisfy the Low Approximate Regret property are ubiquitous and include simple, efficient algorithms such as Hedge and variants. The observation that this broad class of algorithms enjoys fast convergence in realistic settings suggests that fast convergence occurs in practice.

Comparing our work to [28] with regard to feedback, Low Approximate Regret algorithms require only realized feedback, while the analysis of the RVU property in [28] requires expected feedback. To see the contrast, consider the load balancing game introduced in [17] with two players and two bins, where each player selects a bin and observes cost given by the number of players in that bin. Initialized at the uniform distribution, any learning algorithm with expectation feedback (e.g. those in [28]) will stay at the uniform distribution forever, because the expected cost vector distributes

cost equally across the two bins. This gives low regret under expected costs, but suppose we were interested in realized costs: The only "black box" way to lift [28] to this case would be to simply evaluate the regret bound above under realized costs, but here players will experience $\Theta(1/\sqrt{T})$ variation because they select bins uniformly at random, ruining the fast convergence. Our analysis sidesteps this issue because players achieve Low Approximate Regret with high probability.

In Section 4 we consider games where players can only observe the cost of the action they played given the actions taken by the other players, and receive no feedback for actions not played (*bandit feedback*). [22] analyzed zero-sum games with bandit feedback, but assumed that players receive expected cost over the strategies of all other players. In contrast, the Low Approximate Regret property can be satisfied by just observing realizations, even with bandit feedback. We propose a new bandit algorithm based on log-barrier regularization with importance sampling that guarantees fast convergence of $O(d \log T/\epsilon)$ where $d$ is the number of actions. Known techniques would either result in a convergence rate of $O(d^3 \log T)$ (e.g. adaptations of SCRiBLe [21]) or would not extend to utility maximization settings (e.g. GREEN [2]). Our technique is of independent interest since it improves the dependence of approximate regret bounds on the number of experts while applying to both cost minimization and utility maximization settings.

Finally, in Section 5, we consider the *dynamic population game* setting of [19], where players enter and leave the game over time. [19] showed that regret bounds for shifting experts directly influence the rate at which players can turn over and still guarantee close to optimal solutions on average. We show that a number of learning algorithms have the Low Approximate Regret property in the shifting experts setting, allowing us to extend the fast convergence result to dynamic games. Such learning algorithms include a noisy version of Hedge as well as AdaNormalHedge [18], which was previously studied in the dynamic setting in [19]. Low Approximate Regret allows us to increase the turnover rate from the one in [19], while also widening and simplifying the class of learning algorithms that players can use to guarantee the close to optimal average welfare.

## 2 Repeated Games and Learning Dynamics

We consider a game $G$ among a set of $n$ players. Each player $i$ has an action space $S_i$ and a cost function $\mathbf{cost}_i : S_1 \times \cdots \times S_n \to [0,1]$ that maps an action profile $s = (s_1, \ldots, s_n)$ to the cost $\mathbf{cost}_i(s)$ that player experiences[1]. We assume that the action space of each player has cardinality $d$, i.e. $|S_i| = d$. We let $w = (w_1, \ldots, w_n)$ denote a list of probability distributions over all players' actions, where $w_i \in \Delta(S_i)$ and $w_{i,x}$ is the probability of action $x \in S_i$.

The game is repeated for $T$ rounds. At each round $t$ each player $i$ picks a probability distribution $w_i^t \in \Delta(S_i)$ over actions and draws their action $s_i^t$ from this distribution. Depending on the game playing environment under consideration, players will receive different types of feedback after each round. In Sections 3 and 5 we consider feedback where at the end of the round each player $i$ observes the utility they would have received had they played any possible action $x \in S_i$ given the actions taken by the other players. More formally let $c_{i,x}^t = \mathbf{cost}_i(x, s_{-i}^t)$, where $s_{-i}^t$ is the set of strategies of all but the $i^{th}$ player at round $t$, and let $c_i^t = (c_{i,x}^t)_{x \in S_i}$. Note that the expected cost of player $i$ at round $t$ (conditioned on the other players' actions) is simply the inner product $\langle w_i^t, c_i^t \rangle$.

We refer to this form of feedback as *realized feedback* since it only depends on the realized actions $s_{-i}^t$ sampled by the opponents; it does not directly depend on their distributions $w_{-i}^t$. This should be contrasted with the *expectation feedback* used by [28, 9, 22], where player $i$ observes $\mathbb{E}_{s_{-i}^t \sim w_{-i}^t}[\mathbf{cost}_i(x, s_{-i}^t)]$ for each $x$.

Sections 4 and 5 consider extensions of our repeated game model. In Section 4 we examine partial information ("bandit") feedback, where players observe only the cost of their own realized actions. In Section 5 we consider a setting where the player set is evolving over time. Here we use the dynamic population model of [19], where at each round $t$ each player $i$ is replaced ("turns over") with some probability $p$. The new player has cost function $\mathbf{cost}_i^t(\cdot)$ and action space $S_i^t$ which may change arbitrarily subject to certain constraints. We will formalize this notion later on.

**Learning Dynamics** We assume that players select their actions using learning algorithms satisfying a property we call *Low Approximate Regret*, which simply requires that the cumulative cost of the learner multiplicatively approximates the cost of the best action they could have chosen in hindsight.

We will see in subsequent sections that this property is ubiquitous and leads to fast convergence in a robust range of settings.

**Definition 1.** *(Low Approximate Regret) A learning algorithm for player $i$ satisfies the* Low Approximate Regret *property for parameter $\epsilon > 0$ and function $A(d,T)$ if for all action distributions $f \in \Delta(S_i)$,*

$$(1 - \epsilon) \sum_{t=1}^{T} \langle w_i^t, c_i^t \rangle \leq \sum_{t=1}^{T} \langle f, c_i^t \rangle + \frac{A(d,T)}{\epsilon}. \tag{1}$$

*A learning algorithm satisfies Low Approximate Regret against shifting experts if for all sequences $f^1, \ldots, f^T \in \Delta(S_i)$, letting $K = |\{i > 2 : f^{t-1} \neq f^t\}|$ be the number of shifts,*

$$(1 - \epsilon) \sum_{t=1}^{T} \langle w_i^t, c_i^t \rangle \leq \sum_{t=1}^{T} \langle f^t, c_i^t \rangle + (1 + K)\frac{A(d,T)}{\epsilon}. \tag{2}$$

*In the bandit feedback setting, we require (1) to hold in expectation over the realized strategies of player $i$ for any $f \in \Delta(S_i)$ fixed before the game begins.*

We use the version of the Low Approximate Regret property with shifting experts when considering players in dynamic population games in Section 5. In this case, the game environment is constantly changing due to churn in the population, and we need the players to have low approximate regret with shifting experts to guarantee high social welfare despite the churn.

We emphasize that all algorithms we are aware of that satisfy Low Approximate Regret can be made to do so for any fixed choice of the approximation factor $\epsilon$ via an appropriate selection of parameters. Many algorithms have an even stronger property: They satisfy (1) or (2) *for all $\epsilon > 0$ simultaneously*. We say that such algorithms satisfy the *Strong Low Approximate Regret* property. This property has favorable consequences in the context of repeated games.

The Low Approximate Regret property differs from previous properties such as RVU in that it only requires that the learner's cost be close to a multiplicative approximation to the cost of the best action in hindsight. Consequently, it is always smaller than the regret. For instance, if we consider only uniform (i.e. not data-dependent) regret bounds the Hedge algorithm can only achieve $O(\sqrt{T \log d})$ exact regret, but can achieve Low Approximate Regret with parameters $\epsilon$ and $A(d,T) = O(\log d)$ for any $\epsilon > 0$. Low Approximate Regret is analogous to the notion of $\alpha$-regret from [15], with $\alpha = (1 + \epsilon)$.

In Appendix D we show that the Low Approximate Regret property and our subsequent results naturally extend to utility maximization games.

**Smooth Games**   It is well-known that in a large class of games, termed *smooth games* by Roughgarden [23], traditional learning dynamics converge to approximately optimal social welfare. In subsequent sections we analyze the convergence of Low Approximate Regret learning dynamics in such smooth games. We will see that Low Approximate Regret (for sufficiently small $A(d,T)$) coupled with smoothness of the game implies fast convergence of learning dynamics to desirable social welfare under a variety of conditions. Before proving this result we review social welfare and smooth games.

For a given action profile $s$, the social cost is $C(s) = \sum_{i=1}^{n} \mathbf{cost}_i(s)$. To bound the efficiency loss due to the selfish behavior of the players we define

$$\text{OPT} = \min_{s^o} \sum_{i=1}^{n} \mathbf{cost}_i(s^o).$$

**Definition 2.** *(Smooth game [23]) A cost minimization game is called $(\lambda, \mu)$-smooth if for all strategy profiles $s$ and $s^*$: $\sum_i \mathbf{cost}_i(s_i^*, s_{-i}) \leq \lambda \cdot \mathbf{cost}_i(s^*) + \mu \cdot \mathbf{cost}_i(s)$.*

This property is typically applied using a (close to) optimal action profile $s^* = s^o$. For this case the property implies that if $s$ is an action profile with very high cost, then some player deviating to her share of the optimal profile $s_i^*$ will improve her cost.

For smooth games, the price of anarchy is at most $\lambda/(1 - \mu)$, meaning that Nash equilibria of the game, as well as no-regret learning outcomes in the limit, have social cost at most a factor of $\lambda/(1 - \mu)$ above the optimum. Smooth cost minimization games include congestion games such

as routing or load balancing. For example, atomic congestion games with affine cost functions are $(\frac{5}{3}, \frac{1}{3})$-smooth [8], non-atomic games are $(1, 0.25)$ smooth [25], implying a price of anarchy of 2.5 and 1.33 respectively. While we focus on cost-minimization games for simplicity of exposition, an analogous definition also applies for utility maximization, including smooth mechanisms [29], which we elaborate on in Appendix D. Smooth mechanisms include most simple auctions. For example, the first price item auction is $(1 - 1/e, 1)$-smooth and all-pay actions are $(1/2, 1)$-smooth, implying a price of anarchy of 1.58 and 2 respectively. All of our results extend to such mechanisms.

## 3 Learning in Games with Full Information Feedback

We now analyze the efficiency of algorithms with the Low Approximate Regret property in the full information setting. Our first proposition shows that, for smooth games with full information feedback, learners with the Low Approximate Regret property converge to efficient outcomes.

**Proposition 1.** *In any $(\lambda, \mu)$-smooth game, if all players use Low Approximate Regret algorithms satisfying Eq. (1) with parameters $\epsilon$ and $A(d, T)$, then for the action profiles $s^t$ drawn on round $t$ from the corresponding mixed actions of the players,*

$$\frac{1}{T} \sum_t \mathbb{E}\big[C(s^t)\big] \leq \frac{\lambda}{1 - \mu - \epsilon} \text{OPT} + \frac{n}{T} \cdot \frac{1}{1 - \mu - \epsilon} \cdot \frac{A(d, T)}{\epsilon}.$$

**Proof.** This proof is a straightforward modification of the usual price of anarchy proof for smooth games. We obtain the claimed bound by writing $\sum_t \mathbb{E}[C(s^t)] = \sum_i \sum_t \mathbb{E}[\mathbf{cost}_i(s^t)]$, using the Low Approximate Regret property with $f = s_i^*$ for each player $i$ for the optimal solution $s^*$, then using the smoothness property for each time $t$ to bound $\sum_i \mathbf{cost}_i(s_i^*, s_{-i}^t)$, and finally rearranging terms. $\qquad\square$

For $\epsilon << (1 - \mu)$ the approximation factor of $\lambda/(1 - \mu - \epsilon)$ is very close to the price of anarchy $\lambda/(1 - \mu)$. This shows that Low Approximate Regret learning dynamics quickly converge to outcomes with social welfare arbitrarily close to the welfare guaranteed for exact Nash equilibria by the price of anarchy. A simple corollary of this proposition is that, when players use learning algorithms that satisfy the Strong Low Approximate Regret property, the bound above can be taken to depend on OPT even though this value is unknown to the players.

Whenever the Low Approximate Regret property is satisfied, a high probability version of the property with similar dependence on $\epsilon$ and $A(d, T)$ is also satisfied. This implies that in addition to quickly converging to efficient outcomes in expectation, Low Approximate Regret learners experience fast convergence with high probability.

**Proposition 2.** *In any $(\lambda, \mu)$-smooth game, if all players use Low Approximate Regret algorithms satisfying Eq. (1) for parameters $\epsilon$ and $A(d, T)$, then for the action profile $s^t$ drawn on round $t$ from the players' mixed actions and $\gamma = 2\epsilon/(1 + \epsilon)$, we have that $\forall \delta > 0$, with probability at least $1 - \delta$,*

$$\frac{1}{T} \sum_t C(s^t) \leq \frac{\lambda}{1 - \mu - \gamma} \text{OPT} + \frac{n}{T} \cdot \frac{1}{1 - \mu - \gamma} \cdot \left[ \frac{4A(d, T)}{\gamma} + \frac{12 \log(n \log_2(T)/\delta))}{\gamma} \right],$$

**Examples of Simple Low Approximate Regret Algorithms** Propositions 1 and 2 are informative when applied with algorithms for which $A(d, T)$ is sufficiently small. One would hope that such algorithms are relatively simple and easy to find. We show now that the well-known Hedge algorithm as well as basic variants such as Optimistic Hedge and Hedge with online learning rate tuning satisfy the property with $A(d, T) = O(\log d)$, which will lead to fast convergence both in terms of $n$ and $T$. For these algorithms and indeed all that we consider in this paper, we can achieve the Low Approximate Regret property for any fixed $\epsilon > 0$ via an appropriate parameter setting. In Appendix A.2, we provide full descriptions and proofs for these algorithms.

**Example 1.** *Hedge satisfies the Low Approximate Regret property with $A(d, T) = \log(d)$. In particular one can achieve the property for any fixed $\epsilon > 0$ by using $\epsilon$ as the learning rate.*

**Example 2.** *Hedge with online learning rate tuning satisfies the Strong Low Approximate Regret property with $A(d, T) = O(\log d)$.*

**Example 3.** *Optimistic Hedge satisfies the Low Approximate Regret property with $A(d, T) = 8\log(d)$. As with vanilla Hedge, we can choose the learning rate to achieve the property with any $\epsilon$.*

**Example 4.** *Any algorithm satisfying a "small loss" regret bound of the form $\sqrt{(Learner's\ cost)\cdot A}$ or $\sqrt{(Cost\ of\ best\ action)\cdot A}$ satisfies Strong Low Approximate Regret via the AM-GM inequality, i.e. $\sqrt{(Learner's\ cost)\cdot A} \propto \inf_{\epsilon>0}[\epsilon\cdot(Learner's\ cost) + A/\epsilon]$. In particular, this implies that the following algorithms have Strong Low Approximate Regret: Canonical small loss and self-confident algorithms, e.g. [11, 4, 30], Algorithm of [7], Variation MW [13], AEG-Path [26], AdaNormalHedge [18], Squint [16], and Optimistic PAC-Bayes [10].*

Example 4 shows that the Strong Low Approximate Regret property in fact is ubiquitous, as it is satisfied by any algorithm that provides small loss regret bounds or one of many variants on this type of bound. Moreover, all algorithms that satisfy the Low Approximate Regret property for all fixed $\epsilon$ can be made to satisfy the strong property using the doubling trick.

**Main Result for Full Information Games:**

**Theorem 3.** *In any $(\lambda, \mu)$-smooth game, if all players use Low Approximate Regret algorithms satisfying (1) for parameter $\epsilon$ [2] and $A(d, T) = O(\log d)$, then*

$$\frac{1}{T}\sum_t \mathbb{E}[C(s^t)] \le \frac{\lambda}{1-\mu-\epsilon}\mathrm{OPT} + \frac{n}{T}\cdot\frac{1}{1-\mu-\epsilon}\cdot\frac{O(\log d)}{\epsilon},$$

*and furthermore, $\forall \delta > 0$, with probability at least $1 - \delta$,*

$$\frac{1}{T}\sum_t \mathbb{E}[C(s^t)] \le \frac{\lambda}{1-\mu-\epsilon}\mathrm{OPT} + \frac{n}{T}\cdot\frac{1}{1-\mu-\epsilon}\cdot\left[\frac{O(\log d)}{\epsilon} + \frac{O(\log(n\log_2(T)/\delta))}{\epsilon}\right].$$

**Corollary 4.** *If all players use Strong Low Approximate Regret algorithms then: 1. The above results hold for all $\epsilon > 0$ simultaneously. 2. Individual players have regret bounded by $O(T^{-1/2})$, even in adversarial settings. 3. The players approach a coarse correlated equilibrium asymptotically.*

**Comparison with Syrgkanis et al. [28].** By relaxing the standard $\lambda/(1-\mu)$ price of anarchy bound, Theorem 3 substantially broadens the class of algorithms that experience fast convergence to include even the common Hedge algorithm. The main result of [28] shows that learning algorithms that satisfy their RVU property converge to the price of anarchy bound $\lambda/(1-\mu)$ at rate $n^2\log d/T$. They further achieve a worse approximation of $\lambda(1+\mu)/(\mu(1-\mu))$ at the improved (in terms of $n$) rate of $n\log d/T$. We converge to an approximation arbitrarily close to $\lambda/(1+\mu)$ at a rate of $n\log d/T$. Note that in atomic congestion games with affine congestion function $\mu = 1/3$, so their bound of $\lambda(1+\mu)/\mu(1-\mu)$ loses a factor of 4 compared to the price of anarchy.

Strong Low Approximate Regret algorithms such as Hedge with online learning rate tuning simultaneously experience both fast $O(n/T)$ convergence in games and an $O(1/\sqrt{T})$ bound on individual regret in adversarial settings. In contrast, [28] only shows $O(n/\sqrt{T})$ individual regret and $O(n^3/T)$ convergence to price of anarchy simultaneously.

Low Approximate Regret algorithms only need realized feedback, whereas [28] require expectation feedback. Having players receive expectation feedback is unrealistic in terms of both information and computation. Indeed, even if the necessary information was available, computing expectations over discrete probability distributions is not tractable unless $n$ is taken to be constant.

Our results imply that Optimistic Hedge enjoys the best of two worlds: It enjoys fast convergence to the exact $\lambda/(1-\mu)$ price of anarchy using expectation feedback as well as fast convergence to the $\epsilon$-approximate price of anarchy using realized feedback. Our new analysis of Optimistic Hedge (Appendix A.2.2) sheds light on another desirable property of this algorithm: Its regret is bounded in terms of the net cost incurred by Hedge. Figure 1 summarizes the differences between our results.

| | Feedback | POA | Rate | Time comp. |
|---|---|---|---|---|
| RVU property [28] | Expected costs | exact | $O(n^2\log d/T)$ | $d^{O(n)}$ per round |
| LAR property (section 2) | Realized costs | $\epsilon$-approx | $O(n\log d/(\epsilon T))$ | $O(d)$ per round |

Figure 1: Comparison of Low Approximate Regret and RVU properties.

## 4 Bandit Feedback

In many realistic scenarios, the players of a game might not even know what they would have lost or gained if they had deviated from the action they played. We model this lack of information with *bandit feedback*, in which each player observes a single scalar, $\mathbf{cost}_i(s^t) = \langle s_i^t, c_i^t \rangle$, per round.[3] When the game considered is smooth, one can use the Low Approximate Regret property as in the full information setting to show that players quickly converge to efficient outcomes. Our results here hold with the same generality as in the full information setting: As long as learners satisfy the Low Approximate Regret property (1), an efficiency result analogous to Proposition 1 holds.

**Proposition 5.** *Consider a $(\lambda, \mu)$-smooth game. If all players use bandit learning algorithms with Low Approximate Regret $A(d, T)$ then*

$$\frac{1}{T} \mathbb{E}\left[\sum_t C(s^t)\right] \leq \frac{\lambda}{1 - \mu - \epsilon} \mathrm{OPT} + \frac{n}{T} \cdot \frac{1}{1 - \mu - \epsilon} \cdot \frac{A(d, T)}{\epsilon}.$$

**Bandit Algorithms with Low Approximate Regret** The bandit Low Approximate Regret property requires that (1) holds in expectation against any sequence of adaptive and potentially adversarially chosen costs, but only for an obliviously chosen comparator $f$.[4] This is weaker than requiring that an algorithm achieve a true expected regret bound; it is closer to pseudo-regret.

The Exp3Light algorithm [27] satisfies Low Approximate Regret with $A(d, T) = d^2 \log T$. The SCRiBLe algorithm introduced in [1] (via the analysis in [21]) enjoys the Low Approximate Regret property with $A(d, T) = d^3 \log(dT)$. The GREEN algorithm [2] achieves the Low Approximate Regret property with $A(d, T) = d \log(T)$, but only works with costs and not gains. This prevents it from being used in utility settings such as auctions, as in Appendix D.

We present a new bandit algorithm (Algorithm 3) that achieves Low Approximate Regret with $A(d, T) = d \log(T/d)$ and thus matches the performance of GREEN, but works in both cost minimization and utility maximization settings. This method is based on Online Mirror Descent with a logarithmic barrier for the positive orthant, but differs from earlier algorithms based on the logarithmic barrier (e.g. [21]) in that it uses the classical importance-weighted estimator for costs instead of sampling based on the Dikin elipsoid. It can be implemented in $\tilde{O}(d)$ time per round, using line search to find $\gamma$. We provide proofs and further discussion of Algorithm 3 in Appendix B.

**Algorithm 3: Initialize $w^1$ to the uniform distribution. On each round $t$, perform update:**

$$\text{Algorithm 3 update:} \quad \mathbf{w_{s^{t-1}}^t} = \frac{\mathbf{w_{s^{t-1}}^{t-1}}}{1 + \eta \mathbf{c_{s^{t-1}}^t} + \gamma \mathbf{w_{s^{t-1}}^{t-1}}} \quad \text{and} \quad \forall \mathbf{j} \neq \mathbf{s^{t-1}} \ \ \mathbf{w_j^t} = \frac{\mathbf{w_j^{t-1}}}{1 + \gamma \mathbf{w_j^{t-1}}}, \quad (3)$$

**where $\gamma \leq 0$ is chosen so that $w^t$ is a valid probability distribution.**

**Lemma 6.** *Algorithm 3 with $\eta = \epsilon/(1+\epsilon)$ has Low Approximate Regret with $A(d, T) = O(d \log T)$.*

**Comparison to Other Algorithms** In contrast to the full information setting where the most common algorithm, Hedge, achieves Low Approximate Regret with competitive parameters, the most common adversarial bandit algorithm Exp3 does not seem to satisfy Low Approximate Regret. [3] provide a small loss bound for bandits which would be sufficient for Low Approximate Regret, but their algorithm requires prior knowledge on the loss of the best action (or a bound on it), which is not appropriate in our game setting. Similarly, the small loss bound in [20] is not applicable in our setting as the work assumes an oblivious adversary and so does not apply to the games we consider.

## 5 Dynamic Population Games

In this section we consider the dynamic population repeated game setting introduced in [19]. Detailed discussion and proofs are deferred to Appendix C. Given a game $G$ as described in Section 2, a *dynamic population game* with *stage game $G$* is a repeated game where at each round $t$ game $G$ is played and every player $i$ is replaced by a new player with a *turnover probability $p$*. Concretely, when a player turns over, their strategy set and cost function are changed arbitrarily subject to the rules

of the game. This models a repeated game setting where players have to adapt to an adversarially changing environment. We denote the cost function of player $i$ at round $t$ as $\mathbf{cost}_i^t(\cdot)$. As in Section 3, we assume that the players receive full information feedback. At the end of each round they observe the entire cost vector $c_i^t = \mathbf{cost}_i^t(\cdot, s_{-i}^t)$, but are not aware of the costs of other players in the game.

**Learning in Dynamic Population Games and the Price of Anarchy** To guarantee small overall cost using the smoothness analysis from Section 2, players need to exhibit low regret against a shifting benchmark $s_i^{*t}$ of socially optimal strategies achieving $\mathrm{OPT}^t = \min_{s^{*t}} \sum_i \mathbf{cost}_i^t(s^{*t})$. Even with a small probability $p$ of change, the sequence of optimal solutions can have too many changes to be able to achieve low regret. In spite of this apparent difficulty, [19] prove that at least a $\rho\lambda/(1-\mu-\epsilon)$ fraction of the optimal welfare is guaranteed if 1. players are using low *adaptive regret* algorithms (see [14, 18]) and 2. for the underlying optimization problem there exists a relatively *stable* sequence of solutions which at each step approximate the optimal solution by a factor of $\rho$. This holds as long as the turnover probability $p$ is upper bounded by a function of $\epsilon$ (and of certain other properties of the game, such as the stability of the close to optimal solution).

We consider dynamic population games where each player uses a learning algorithm satisfying Low Approximate Regret for shifting experts (2). This shifting version of Low Approximate Regret implies a dynamic game analog of our main efficiency result, Proposition 1.

**Algorithms with Low Approximate Regret for Shifting Experts** A simple variant of Hedge we term *Noisy Hedge*, which mixes the Hedge update at each round with a small amount of uniform noise, satisfies the Low Approximate Regret property for shifting experts with $A(d, T) = O(\log(dT))$. Moreover, algorithms that satisfy a small loss version of the *adaptive regret* property [14] used in [19] satisfy the Strong Low Approximate Regret property.

**Proposition 7.** *Noisy Hedge with learning rate $\eta = \epsilon$ satisfies the Low Approximate Regret property for shifting experts with $A(d, T) = 2\log(dT)$.*

**Extending Proposition 1 to the Dynamic Population Game Setting** Let $s^{*1:T}$ denote a stable sequence of near-optimal solutions $s^{*t}$ with $\sum_i \mathbf{cost}_i^t(s^{*t}) \leq \rho \cdot \mathrm{OPT}^t$ for all rounds $t$. As discussed in [19], such stable sequences can come from simple greedy algorithms (where each change in the input of one player affects the output of few other players) or via differentially private algorithms (where each change in the input of one player affects the output of all other players with small probability); in the latter case the sequence is randomized. For a deterministic sequence $s_i^{*1:T}$ of player $i$'s actions, we let the random variable $K_i$ denote the number of changes in the sequence. For a randomized sequence $s_i^{*1:T}$, we let $K_i$ be the sum of total variation distances between subsequent pairs $s_i^{*t-1}$ and $s_i^{*t}$. The stability of a sequence of solutions is determined by $\mathbb{E}[\sum_i K_i]$.

**Proposition 8.** *(PoA with Dynamic Population) If all players use Low Approximate Regret algorithms satisfying (2) in a dynamic population game, where the stage game is $(\lambda, \mu)$-smooth, and $K_i$ as defined above then*

$$\frac{1}{T}\sum_t \mathbb{E}[C(s^t)] \leq \frac{1}{T}\frac{\lambda \cdot \rho}{1-\mu-\epsilon}\sum_t \mathbb{E}[\mathrm{OPT}^t] + \frac{n + \mathbb{E}[\sum_i K_i]}{T} \cdot \frac{1}{1-\mu-\epsilon} \cdot \frac{A(d,T)}{\epsilon}. \quad (4)$$

*Here the expectation is taken over the random turnover in the population playing the game, as well as the random choices of the players on the left hand side.*

To claim a price of anarchy bound, we need to ensure that the additive term in (4) is a small fraction of the optimal cost. The challenge is that high turnover probability reduces stability, increasing $\mathbb{E}[\sum_i K_i]$. By using algorithms with smaller $A(d, T)$, we can allow for higher $\mathbb{E}[\sum_i K_i]$ and hence higher turnover probability. Combining Noisy Hedge with Proposition 8 strengthens the results in [19] by both weakening the behavioral assumption on the players, allowing them to use simpler learning algorithms, and allowing a higher turnover probability.

**Comparison to Previous Results** [19] use the more complex AdaNormalHedge algorithm of [18], which satisfies the adaptive regret property of [14], but has $O(dT)$ space complexity. In contrast, Noisy Hedge only requires space complexity of just $O(d)$. Moreover, a broader class of algorithms satisfy the Low Approximate Regret property which makes the efficiency guarantees more prescriptive since this property serves as a behavioral assumption. Finally, the our guarantees we provide improve on the turnover probability that can be accommodated as discussed in Appendix C.1.

**Acknowledgements** We thank Vasilis Syrgkanis for sharing his simulation software and the NIPS reviewers for pointing out the GREEN algorithm [2].

## Footnotes

[1]See Appendix D for analogous definitions for utility maximization games.

[2]We can also show that the theorem holds if players satisfy the property for different values of $\epsilon$, but with a dependence on the worst case value of $\epsilon$ across all players.

[3]With slight abuse of notation, $s_i^t$ denotes the identity vector associated to the strategy player $i$ used at time $t$.

[4]This is because we only need to evaluate (1) with the game's optimal solution $s^\star$ to prove efficiency results.

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
