[Supplementary Material]

# APPENDIX

## A   Supplementary material for Section 3

### A.1   Proof of Proposition 2

**Overview**   The core of Proposition 2 is Lemma 9, which shows that the regret of an individual player concentrates around its expectation. To prove the high probability efficiency result of Proposition 2 we simply apply this lemma to the individual players, apply the union bound to get a regret statement that holds for all players simultaneously, then finally apply the same smoothness argument used in the expectation case.

**Lemma 9** (High-probability regret bound). *Let $w^t \in \Delta(d)$ be selected by an algorithm satisfying the Low Approximate Regret property (1) for $\epsilon > 0$ given costs $c^t$ selected by an adaptive adversary, and let $s^t \sim w^t$ be the algorithm's realized action. Then for all $\delta \in (0, \min\{1, n \log_2 T/e\})$ and $T \geq 4$, with probability at least $1 - \delta$,*

$$(1 - \gamma) \sum_{t=1}^{T} \langle s^t, c^t \rangle \leq \sum_{t=1}^{T} \langle s^\star, c^t \rangle + \frac{4A(d,T)}{\gamma} + \frac{12 \log(\log_2(T)/\delta)}{\gamma}, \tag{5}$$

*where $\epsilon = \gamma/(2 - \gamma)$.*

Before proving Lemma 9 we restate a refinement of Freedman's martingale Bernstein inequality due to [5] which is a standard tool for proving high-probability versions of data-dependent regret bounds.

**Lemma 10** ([5]). *Let $X_1, \ldots, X_T$ be a martingale difference sequence with $|X_t| \leq b$. Let $\bar{\sigma}^2 = \sum_{t=1}^{T} Var(X_t \mid X_1, \ldots, X_{t-1})$ be the sum of conditional variances for a particular outcome $X_1, \ldots, X_T$. For all $\delta \in (0, 1/e)$, $T \geq 4$ we have*

$$\mathbb{P}\left( \sum_{t=1}^{T} X_t > 4\sqrt{\bar{\sigma}^2 \log(1/\delta)} + 2b \log(1/\delta) \right) \leq \log_2(T)\delta. \tag{6}$$

**Proof of Lemma 9.** Let $Z^t(s^1, \ldots, s^t) = (1 - \epsilon)\langle s^t, c^t \rangle$ be a random process indexed by $t \in [T]$. We leave the dependence of $c^t$ on $s^1, \ldots, s^{t-1}$ implicit. Let $X^t(s^1, \ldots, s^t) = Z^t(s^1, \ldots, s^t) - \mathbb{E}[Z^t \mid s^1, \ldots, s^{t-1}]$ be the associated martingale difference sequence. Note that $|X^t| \leq (1 - \epsilon) \leq 1$ and that $\mathbb{E}[s^t \mid s^1, \ldots, s^{t-1}] = w^t$.

Lemma 10 applied to $\sum_{t=1}^{T} X^t(s^1, \ldots, s^t)$ and the Low Approximate Regret property (1) now imply that for a given draw of $s^1, \ldots, s^T$, with probability at least $1 - \delta$,

$$(1 - \epsilon) \sum_{t=1}^{T} \langle s^t, c^t \rangle = \sum_{t=1}^{T} Z_t \leq \sum_{t=1}^{T} \mathbb{E}[Z^t \mid s^1, \ldots, s^{t-1}] + 4\sqrt{\bar{\sigma}^2 \log(\log_2(T)/\delta)} + 2 \log(\log_2(T)/\delta)$$

$$= (1 - \epsilon) \sum_{t=1}^{T} \langle w^t, c^t \rangle + 4\sqrt{\bar{\sigma}^2 \log(\log_2(T)/\delta)} + 2 \log(\log_2(T)/\delta)$$

$$\leq \sum_{t=1}^{T} \langle s^\star, c^t \rangle + \frac{A(d,T)}{\epsilon} + 4\sqrt{\bar{\sigma}^2 \log(\log_2(T)/\delta)} + 2 \log(\log_2(T)/\delta),$$

where $s^\star = \arg\min_{i \in [d]} \sum_{t=1}^{T} \langle e_i, c^t \rangle$. To complete this bound we must provide a bound on the conditional variance $\bar{\sigma}^2$. To this end note that

$$\bar{\sigma}^2 = \sum_{t=1}^{T} \mathbb{E}\left[ \left(X_s^t\right)^2 \mid s^1, \ldots, s^{t-1} \right]$$

$$= (1 - \epsilon)^2 \sum_{t=1}^{T} \mathbb{E}\left[ \left(\langle s^t, c^t \rangle - \langle w^t, c^t \rangle\right)^2 \mid s^1, \ldots, s^{t-1} \right].$$

Now, since the mean minimizes the squared error:

$$\leq (1-\epsilon)^2 \sum_{t=1}^{T} \mathbb{E}\Big[\big(\langle s^t, c^t\rangle\big)^2 \mid s^1, \dots, s^{t-1}\Big].$$

Since $c^t \in [0,1]^d$ we have

$$\bar{\sigma}^2 \leq (1-\epsilon)^2 \sum_{t=1}^{T} \mathbb{E}\big[\langle s^t, c^t\rangle \mid s^1, \dots, s^{t-1}\big]$$

$$= (1-\epsilon)^2 \sum_{t=1}^{T} \langle w^t, c^t\rangle.$$

Hence, with probability at least $1-\delta$,

$$(1-\epsilon) \sum_{t=1}^{T} \langle s^t, c^t\rangle - \sum_{t=1}^{T} \langle s^\star, c^t\rangle$$

$$\leq \frac{A(d,T)}{\epsilon} + 4\sqrt{(1-\epsilon)^2 \left(\sum_{t=1}^{T} \langle w^t, c^t\rangle\right) \log(\log_2(T)/\delta) + 2\log(\log_2(T)/\delta)}.$$

Now for all $\epsilon' > 0$ by the AM-GM inequality we have:

$$\leq \frac{A(d,T)}{\epsilon} + \epsilon'(1-\epsilon)^2 \left(\sum_{t=1}^{T} \langle w^t, c^t\rangle\right) + 4\log(\log_2(T)/\delta)/\epsilon' + 2\log(\log_2(T)/\delta),$$

and so the Low Approximate Regret property (1) implies

$$\leq \frac{A(d,T)}{\epsilon} + \epsilon'(1-\epsilon)\left[\sum_{t=1}^{T} \langle s^\star, c^t\rangle + \frac{A(d,T)}{\epsilon}\right] + 4\log(\log_2(T)/\delta)/\epsilon' + 2\log(\log_2(T)/\delta).$$

Rearranging,

$$\frac{(1-\epsilon)}{(1+\epsilon'(1-\epsilon))} \sum_{t=1}^{T} \langle s^t, c^t\rangle$$

$$\leq \sum_{t=1}^{T} \langle s^\star, c^t\rangle + \frac{1}{1+\epsilon'(1-\epsilon)}\left[\frac{A(d,T)}{\epsilon} + 4\log(\log_2(T)/\delta)/\epsilon' + 2\log(\log_2(T)/\delta)\right] + \frac{A(d,T)}{\epsilon}.$$

Taking $\epsilon' = \epsilon$ we have

$$\frac{(1-\epsilon)}{1+\epsilon-\epsilon^2} \sum_{t=1}^{T} \langle s^t, c^t\rangle \leq \sum_{t=1}^{T} \langle s^\star, c^t\rangle + 2\frac{A(d,T)}{\epsilon} + \frac{6\log(\log_2(T)/\delta)}{\epsilon}.$$

We simplify to a slightly weaker bound,

$$\frac{(1-\epsilon)}{(1+\epsilon)} \sum_{t=1}^{T} \langle s^t, c^t\rangle \leq \sum_{t=1}^{T} \langle s^\star, c^t\rangle + 2\frac{A(d,T)}{\epsilon} + \frac{6\log(\log_2(T)/\delta)}{\epsilon}.$$

Now setting $\epsilon = \gamma/(1-\gamma)$ we arrive at

$$(1-2\gamma) \sum_{t=1}^{T} \langle s^t, c^t\rangle \leq \sum_{t=1}^{T} \langle s^\star, c^t\rangle + 2\frac{A(d,T)}{\gamma} + \frac{6\log(\log_2(T)/\delta)}{\gamma}.$$

Finally, reparameterizing with $\gamma' = 2\gamma$ we have

$$(1-\gamma') \sum_{t=1}^{T} \langle s^t, c^t\rangle \leq \sum_{t=1}^{T} \langle s^\star, c^t\rangle + \frac{4A(d,T)}{\gamma'} + \frac{12\log(\log_2(T)/\delta)}{\gamma'}.$$

$\square$

## A.2 Low Approximate Property for Specific Algorithms

In this section, we present the proofs of the Low Approximate Regret property for Hedge (Example 1) and Optimistic Hedge (Example 3). The first proof is only included for completeness but may be helpful as subsequent proofs follow the same framework. Our proof for Optimistic Hedge includes a new analysis that relates the performance of Optimistic Hedge on a given cost sequence to the performance of Hedge on the same sequence. The analysis shows that Optimistic Hedge will experience low regret whenever Hedge has low cost, which in particular implies that it satisfies the Low Approximate Regret property. We omit the proof for Example 2 and instead refer the reader to Corollary 2.4 of [6], which derives the result using the doubling trick.

### A.2.1 Hedge (Example 1)

Hedge is an algorithm for online linear optimization over the simplex $\Delta(d)$. It has update rule

$$w_i^{t+1} \propto w_i^t e^{-\eta c_i^t} \quad \forall i \in [d],$$

where $\eta > 0$ is the *learning rate*.

We derive Hedge as an instance of Online Mirror Descent (see e.g. [12]) with the *negative entropy regularizer* $R(w) = \sum_{i=1}^d w_i \log(w_i)$. To run Online Mirror Descent one picks a learning rate $\eta > 0$ and initial weights (also known as a *prior*) $w^1$, then performs the following update step at each time $t \in [T]$:

1. Let $\widetilde{w}^t$ satisfy $\nabla R(\widetilde{w}^{t+1}) = \nabla R(w^t) - \eta c^t$.
2. $w^{t+1} = \arg\min_{f \in \Delta(d)} D_R(f|\widetilde{w}^{t+1})$.

Here $D_R(f|g) \triangleq R(f) - R(g) - \langle \nabla R(g), f - g \rangle$ is the *Bregman divergence* for the regularizer $R$. We briefly restate some useful properties of the Mirror Descent update.

**Lemma 11** (Properties of Mirror Descent (e.g. [12])). *For any convex regularizer $R$ we have*

- $D_R(f \mid g) \geq 0$.
- *For any $a, b, c \in \Delta(d)$,*

$$\langle b - a, \nabla R(c) - \nabla R(b) \rangle = D_R(a \mid c) - D_R(a \mid b) - D_R(b \mid c).$$

- *The Mirror Descent update can alternatively be expressed as*

$$w^{t+1} = \arg\min_{f \in \Delta(d)} \eta \langle f, c^t \rangle + D_R(f \mid w^t).$$

- *Any update of the form $f^* = \arg\min_{f \in \Delta(d)} \langle f, c \rangle + D_R(f \mid w)$ satisfies*

$$\langle f^* - g, c \rangle \leq D_R(g \mid w) - D_R(g \mid f^*) - D_R(f^* \mid w) \quad \forall g \in \Delta(d).$$

**Proposition 12.** *Hedge, when run with constant learning rate and uniform prior $w_i^1 = 1/d$, satisfies the Low Approximate Regret property with $A(d, T) = \log(d)$.*

**Proof of Proposition 12.** Using the standard Online Mirror Descent analysis we have that at every step $t$, for any $f \in \Delta(d)$:

$$\langle w^t - f, c^t \rangle \leq \langle w^t - \widetilde{w}^{t+1}, c^t \rangle + \frac{1}{\eta} \left( D_R(f|w^t) - D_R(f|w^{t+1}) - D_R(\widetilde{w}^{t+1}|w^t) \right)$$

$$\leq \langle w^t - \widetilde{w}^{t+1}, c^t \rangle + \frac{1}{\eta} \left( D_R(f|w^t) - D_R(f|w^{t+1}) \right). \tag{7}$$

For the first term in the sum above, we have:

$$\langle w^t - \widetilde{w}^{t+1}, c^t \rangle \leq \eta \langle w^t, c^t \rangle \tag{8}$$

To see this note that $\nabla R(w) = \log(w) + 1$ (where $\log$ is applied element-wise) and hence $(\nabla R)^{-1}(f) = e^{f-1}$. This implies $\widetilde{w}_i^{t+1} = w_i^t e^{-\eta c_i^t}$, and so

$$\langle w^t - \widetilde{w}^{t+1}, c^t \rangle = \sum_{j \in [d]} w_j^t c_j^t (1 - e^{-\eta c_j^t}) \leq \eta \sum_{j \in [d]} w_j^t (c_j^t)^2 \leq \eta \langle w^t, c^t \rangle. \tag{9}$$

The first inequality in (9) uses that $1 - e^{-\eta x} \leq \eta x$ for $x > 0$ and the second inequality uses that the losses lie in $[0, 1]$.

Using relations (7) and (8), and summing over $t$:

$$\sum_t \langle w^t - f, c^t \rangle \leq \eta \sum_t \langle w^t, c^t \rangle + \frac{1}{\eta} D_R(f|w^1). \tag{10}$$

Since $w^1$ is the uniform distribution, $D_R(f|w^1) \leq \log(d)$. Rearranging yields the claimed result. $\square$

### A.2.2 Optimistic Hedge (Example 3)

The Optimistic Hedge algorithm performs two separate weight updates at each timestep to produce its action distribution. The method first performs a Hedge update $g_i^{t+1} \propto g_i^t e^{-\eta c_i^t}$, then produces the prediction distribution: $w_i^{t+1} \propto g_i^{t+1} e^{-\eta c_i^t}$.

**Lemma 13.** *Optimistic Hedge with a constant learning rate $\eta = \epsilon/8 < 1/4$ satisfies the Low Approximate Regret property with $A(d, T) = 8 \log(d)$.*

Let $R$ be the negative entropy regularizer as in the proof of Proposition 12. Let $\nabla^2 R(w)$ denote the Hessian of the regularizer $R$. The *local norm* with respect to $w$ is $\|f\|_w = \sqrt{f^T \nabla^2 R(w) f}$ and its dual norm is $\|x\|_w^\star = \sqrt{x^T (\nabla^2 R(w))^{-1} x}$. For the negative entropy regularizer this definition yields $\|f\|_w^2 = \sum_{i \in [d]} \frac{(f_i)^2}{w_i}$ and $(\|x\|_w^\star)^2 = \sum_{i \in [d]} w_i(x_i)^2$. We begin by restating an intermediate Lemma from [21] that bounds the regret of Optimistic Hedge in terms of the local norm.

**Lemma 14.** *(Lemma 3 in [21]) Optimistic Hedge enjoys for any $f \in \Delta(S)$*

$$\sum_{t=1}^{T} \langle w^t - f, c^t \rangle \leq 2\eta \sum_{t=1}^{T} (\|c^t - c^{t-1}\|_{w^t}^\star)^2 + \frac{\log(d)}{\eta}. \tag{11}$$

*as long as $\eta \|c^t - c^{t-1}\|_\infty \leq 1/4$ at every step.*

**Proof of Lemma 13.** We will focus on the first term in the right-hand side of (11) and prove that for all $t$,

$$(\|c^t - c^{t-1}\|_{w^t}^\star)^2 \leq 2\langle w^t, c^t \rangle + 2\langle g^{t-1}, c^{t-1} \rangle. \tag{12}$$

This holds as

$$(\|c^t - c^{t-1}\|_{w^t}^\star)^2 \leq 2\big((\|c^t\|_{w^t}^\star)^2 + (\|c^{t-1}\|_{w^t}^\star)^2\big)$$

$$= 2\big(\sum_{j=1}^{d} w_j^t (c_j^t)^2 + \sum_{j=1}^{d} w_j^t (c_j^{t-1})^2\big) \tag{13}$$

$$\leq 2\big(\langle w^t, c^t \rangle + \langle w^t, c^{t-1} \rangle\big) \tag{14}$$

$$= 2\big(\langle w^t, c^t \rangle + \langle g^{t-1}, c^{t-1} \rangle + \langle w^t - g^t, c^{t-1} \rangle + \langle g^t - g^{t-1}, c^{t-1} \rangle\big)$$

$$\leq 2\big(\langle w^t, c^t \rangle + \langle g^{t-1}, c^{t-1} \rangle\big). \tag{15}$$

Here (13) holds by the definition of the local norm, (14) holds as the costs are in $[0, 1]$, and (15) holds via two applications of Lemma 11 for Bregman projections:

$$\langle w^t - g^t, c^{t-1} \rangle \leq D_R(g^t \mid g^t) - D_R(g^t \mid w^t) - D_R(w^t \mid g^t) \leq 0.$$

$$\langle g^t - g^{t-1}, c^{t-1} \rangle \leq D_R(g^{t-1} \mid g^{t-1}) - D_R(g^{t-1} \mid g^t) - D_R(g^t \mid g^{t-1}) \leq 0.$$

Now, applying (12) to Lemma 14, we have that for $\eta < 1/4$,

$$\sum_{t=1}^{T} \langle w^t - f, c^t \rangle \leq 4\eta \sum_{t=1}^{T} \langle w^t, c^t \rangle + 4\eta \sum_{t=1}^{T} \langle g^{t-1}, c^{t-1} \rangle + \frac{\log(d)}{\eta}. \tag{16}$$

Observe now that $g^t$ are the weights selected by the basic Hedge algorithm on the sequence $\{c^t\}$ (setting $c^0 = 0$ and $g^0$ uniform). Hence by the Low Approximate Regret property for Hedge (Example 1) we have

$$\sum_{t=1}^{T}\langle w^t - f, c^t\rangle \leq 4\eta\sum_{t=1}^{T}\langle w^t, c^t\rangle + \frac{4\eta}{1-\eta}\Big(\sum_{t=1}^{T}\langle f, c^{t-1}\rangle + \frac{\log(d)}{\eta}\Big) + \frac{\log(d)}{\eta}.$$

Rearranging,

$$(1-4\eta)\sum_{t=1}^{T}\langle w^t, c^t\rangle \leq \frac{1+3\eta}{1-\eta}\Big(\sum_{t=1}^{T}\langle f, c^t\rangle + \frac{\log(d)}{\eta}\Big).$$

This gives the claimed bound as $1+3\eta \leq \frac{1}{1-3\eta}$ for $\eta \leq 1/3$ and $1-8\eta \leq (1-4\eta)(1-\eta)(1-3\eta)$. □

### A.3 Proof of Theorem 3

Theorem 3 follows immediately from Propositions 1 and 2.

Corrollary 4 holds because the Strong Low Approximate Regret property states that (1) holds for all $\epsilon > 0$, so in particular we can set $\epsilon = \sqrt{\frac{\log(d)}{T}}$ to arrive at the desired regret bound.

## B   Supplementary material for Section 4

### B.1   Proof of Lemma 6

Algorithm 3 follows a standard design scheme for bandit algorithms. First we develop an algorithm with a full information regret bound, then run this algorithm using an unbiased estimator for the cost.

Let $R(w) = \sum_{j\in[d]}\log(1/w_j)$; we call this the log barrier regularizer because it is a logarithmic barrier for the positive orthant. Algorithm 3 is equivalent to the following update step at each time $t$:

- Sample $s^t \sim w^t$.
- Observe $c^t_{s^t}$ and build the importance-weighted estimator:

$$\hat{c}^t_j = \begin{cases} c^t_j/w^t_j & \text{if } j = s^t \\ 0 & \text{otherwise} \end{cases}.$$

- Update $w^{t+1}$ with a Mirror Descent step from $\hat{c}^t$:
  1. Let $\widetilde{w}^t$ satisfy $\nabla R(\widetilde{w}^{t+1}) = \nabla R(w^t) - \eta\hat{c}^t$.
  2. $w^{t+1} = \arg\min_{f\in\Delta(d)} D_R(f|\widetilde{w}^{t+1})$.

Note that $\hat{c}^t$ is *unbiased* in that it satisfies $\mathbb{E}_{s^t\sim w^t}[\hat{c}^t] = c^t$.

**Overview**   In this section we state and prove Lemma 17, which provides a regret bound for Algorithm 3. To prove Lemma 6, we apply Lemma 17 with learning rate $\eta = \epsilon/(1+\epsilon)$ and observe that the Low Approximate Regret property is satisfied:

$$(1-\epsilon)\,\mathbb{E}\left[\sum_t\langle e_{s^t}, c^t\rangle\right] \leq \sum_t\langle f, c^t\rangle + \frac{d(1+\epsilon)\log(T/d)}{\epsilon} + d.$$

In Appendix D we sketch a proof of the regret bound for Algorithm 3 in the case where utilities are used instead of costs.

**Regret Bound for Full Information**

**Proposition 15** (Properties of the log barrier regularizer). *Recall that $R(w) = \sum_{i\in[d]}\log(1/w_i)$.*

- $\nabla R(w) = -1/w$, *which implies that for all $i$:* $\widetilde{w}^{t+1}_i = \frac{w^t_i}{1+\eta w^t_i \hat{c}^t_i}$.

- $D_R(f \mid w) = \sum_{i \in [d]} \left[ \log\left(\frac{w_i}{f_i}\right) + \frac{f_i}{w_i} \right] - d.$

**Lemma 16.** *Online Mirror Descent (see section A.2.1) with the log barrier regularizer, for any sequence of costs $c^1, \ldots, c^T$ in $\mathbb{R}^d$, produces weights $w^t$ that satisfy the following bound for any $f^\star \in \Delta(d)$:*

$$\langle w^t - f^\star, c^t \rangle \leq \eta \sum_{j \in [d]} \frac{(w_j^t \cdot c_j^t)^2}{1 + \eta w_j^t c_j^t} + \frac{1}{\eta}\left(D_R(f^\star \mid w^t) - D_R(f^\star \mid w^{t+1})\right). \tag{17}$$

*In particular, it achieves the regret bound*

$$\sum_{t=1}^{T} \langle w^t - f^\star, c^t \rangle \leq \eta \sum_{t=1}^{T} \sum_{j \in [d]} \frac{(w_j^t \cdot c_j^t)^2}{1 + \eta w_j^t c_j^t} + \frac{1}{\eta} D_R(f^\star \mid w^1). \tag{18}$$

**Proof of Lemma 16.** Fix $f^\star \in \Delta(d)$. Starting from the standard Mirror Descent proof we have that for each $t$:

$$\langle w^t - f^\star, \hat{c}^t \rangle \leq \langle w^t - \widetilde{w}^{t+1}, \hat{c}^t \rangle + \frac{1}{\eta}\left(D_R(f^\star \mid w^t) - D_R(f^\star \mid w^{t+1})\right).$$

The result is obtained by plugging in the expression for $\widetilde{w}^t$ from Proposition 15:

$$= \eta \sum_j \frac{(w_j^t \hat{c}_j^t)^2}{1 + \eta w_j^t \hat{c}_j^t} + \frac{1}{\eta}\left(D_R(f^\star \mid w^t) - D_R(f^\star \mid w^{t+1})\right).$$

The regret bound is obtained by summing this inequality. $\qquad \square$

**From Full Information to Partial Information**

**Lemma 17** (Regret bound for Algorithm 3). *For any $f^\star \in \Delta(d)$, and any sequence of costs $c^1, \ldots, c^T \in [0,1]^d$, the weights generated by Algorithm 3 with $\eta \in (0,1)$ satisfy*

$$\mathbb{E}\left[\sum_{t=1}^{T} \langle w^t - f^\star, c^t \rangle\right] \leq \frac{\eta}{1 - \eta} \mathbb{E}\left[\sum_{t=1}^{T} \langle w^t, c^t \rangle\right] + \frac{1}{\eta} d \log(T/d) + d. \tag{19}$$

**Proof of Lemma 17.** Observe that Algorithm 3 is equivalent to running Online Mirror Descent with $R$, using the unbiased estimator $\hat{c}^t$ for costs, where we recall $\hat{c}_i^t = \mathbb{1}\{s^t = i\} c_i^t / w_i^t$. Thus, Lemma 16 implies that at each time $t$,

$$\langle w^t - f^\star, \hat{c}^t \rangle \leq \eta \sum_{j \in [d]} \frac{(w_j^t \cdot \hat{c}_j^t)^2}{1 + \eta w_j^t \hat{c}_j^t} + \frac{1}{\eta}\left(D_R(f^\star \mid w^t) - D_R(f^\star \mid w^{t+1})\right).$$

Since $w_j^t \hat{c}_j^t = \mathbb{1}\{s^t = j\} c_j^t$, we have:

$$= \eta \sum_{j \in [d]} \mathbb{1}\{s^t = j\} \frac{(c_j^t)^2}{1 + \eta c_j^t} + \frac{1}{\eta}\left(D_R(f^\star \mid w^t) - D_R(f^\star \mid w^{t+1})\right)$$

Taking the conditional expectation of each side of this inequality we have

$$\langle w^t - f^\star, c^t \rangle = \mathbb{E}\left[\langle w^t - f^\star, \hat{c}^t \rangle \mid s^1, \ldots, s^{t-1}\right]$$

$$\leq \mathbb{E}\left[\eta \sum_{j \in [d]} \mathbb{1}\{s^t = j\} \frac{(c_j^t)^2}{1 + \eta c_j^t} \mid s^1, \ldots, s^{t-1}\right] + \frac{1}{\eta}\left(D_R(f^\star \mid w^t) - D_R(f^\star \mid w^{t+1})\right)$$

$$= \eta \sum_{j \in [d]} \frac{w_j^t (c_j^t)^2}{1 + \eta c_j^t} + \frac{1}{\eta}\left(D_R(f^\star \mid w^t) - D_R(f^\star \mid w^{t+1})\right).$$

Now, since $c_j^t$ lie in the range $[0,1]$ we have

$$\leq \frac{\eta}{1 - \eta} \langle w^t, c^t \rangle + \frac{1}{\eta}\left(D_R(f^\star \mid w^t) - D_R(f^\star \mid w^{t+1})\right).$$

Summing over all $t$ and taking a final expectation yields the bound,

$$\mathbb{E}\left[\sum_{t=1}^{T}\langle w^t - f^\star, c^t\rangle\right] \le \frac{\eta}{1-\eta}\mathbb{E}\left[\sum_{t=1}^{T}\langle w^t, c^t\rangle\right] + \frac{1}{\eta}D_R(f^\star \mid w^1). \tag{20}$$

It remains to bound the Bregman divergence term. A direct approach fails here because one can choose $f^\star$ to make $D_R(f^\star \mid w^1)$ arbitrarily large; this is in contrast with the case where $D_R$ is the KL divergence, where we have a $\log d$ bound as long as $w^1$ is uniform. To sidestep this difficulty, given arbitrary $f^\star \in \Delta(d)$ we let $\bar{f} = (1-\theta)f^\star + \theta\pi$, where $\theta \in [0,1]$ and $\pi$ is the uniform distribution. By Proposition 15 we have

$$D_R(\bar{f} \mid w^1) \le d\log(1/\theta) \tag{21}$$

Applying (20) with $\bar{f}$ as the comparator now implies

$$\mathbb{E}\left[\sum_{t=1}^{T}\langle w^t - \bar{f}, c^t\rangle\right] \le \frac{\eta}{1-\eta}\mathbb{E}\left[\sum_{t=1}^{T}\langle w^t, c^t\rangle\right] + \frac{1}{\eta}d\log(1/\theta).$$

Rearranging, this implies

$$\mathbb{E}\left[\sum_{t=1}^{T}\langle w^t - f^\star, c^t\rangle\right] \le \frac{\eta}{1-\eta}\mathbb{E}\left[\sum_{t=1}^{T}\langle w^t, c^t\rangle\right] + \theta\sum_{t=1}^{T}\langle \pi, c^t\rangle + \frac{1}{\eta}d\log(1/\theta).$$

Since we have assumed $c^t \in [0,1]^d$, this is bounded as

$$\le \frac{\eta}{1-\eta}\mathbb{E}\left[\sum_{t=1}^{T}\langle w^t, c^t\rangle\right] + \theta T + \frac{1}{\eta}d\log(1/\theta).$$

Finally, setting $\theta = d/T$ yields the desired bound:

$$\le \frac{\eta}{1-\eta}\mathbb{E}\left[\sum_{t=1}^{T}\langle w^t, c^t\rangle\right] + \frac{1}{\eta}d\log(T/d) + d.$$

$\square$

## C  Supplementary Material for Section 5

### C.1  Discussion of Results for Dynamic Population Games

We briefly show how Proposition 8 with players using Low Approximate Regret algorithms with $A(d,T) = O(\log(dT))$ improves the maximum turnover rate $p$ in the results of [19].

In Definition 1, Low Approximate Regret for shifting experts (2) is defined in terms of the number of shifts $K = |\{i > 2 : f^{t-1} \neq f^t\}|$ in a sequence of comparators $f^1, \ldots, f^T$. To compare with [19] we need a slightly different notion of Low Approximate Regret based on the *total variation distance* of the sequence $f^1, \ldots, f^T$. Letting $K = \sum_t \lVert f^t - f^{t-1} \rVert_1$, we require

$$(1-\epsilon)\sum_{t=1}^{T}\langle w_i^t, c_i^t\rangle \le \sum_{t=1}^{T}\langle f^t, c_i^t\rangle + (1+K)\frac{A(d,T)}{\epsilon}. \tag{22}$$

In fact, whenever Low Approximate Regret for shifting experts (2) holds, (22) holds as well as explained in [19]. Thus, without loss of generality we take $K_i$ to be the total variation distance of the solution sequence $s_i^{*1:T}$ for the $i$th player going forward, since if player $i$ satisfies Low Approximate Regret for shifting experts (2) they also satisfy:

$$(1-\epsilon)\sum_{t=1}^{T}\langle w_i^t, c_i^t\rangle \le \sum_{t=1}^{T}\langle s_i^{*t}, c_i^t\rangle + (1 + \sum_{t=2}^{T}\lVert s_i^{*t} - s_i^{*t-1}\rVert_1)\frac{A(d,T)}{\epsilon}. \tag{23}$$

Let $\kappa$ denote the expected number of players whose strategy in $s^{*1:T}$ changes as one player turns over, so $\mathbb{E}[\sum_i K_i] = pnT\kappa$ (as in expectation $pn$ players turn over at each step). The parameter $\kappa$ as defined here depends on the concrete game; it is a parameter of a high stability approximate optimization method used as in [19]. Let $\gamma > 0$ be a lower bound on the minimum cost of each player, at each time step, so that we have $\sum_t \mathbb{E}[\text{OPT}^t] \geq \gamma nT$. Using the two parameters $\kappa$ and $\gamma$, [19] show a price of anarchy bound of $\lambda\rho/(1-\mu-\epsilon)$, assuming the turnover probability $p$ satsifies $p \leq \epsilon^2\gamma^2/(\kappa\log(dT))$. Using Proposition 8 with with $A(d,T) = O(\log(dT))$ (as in, for example, Noisy Hedge) we get the same price of anarchy bound, yet allow higher turnover probability by a factor of $1/\gamma$: We tolerate $p \leq \epsilon^2\gamma/(\kappa\log(dT))$.

To illustrate this improvement, consider matching markets. Suppose $n$ players are each bidding in a first price item auction for one of many items (i.e., the winner pays her own bid for each item). Further suppose $v_{ij}$, the player $i$'s value for item $j$, has $v_{ij} \in [\gamma, 1]$, and that the players are unit-demand, each bidding with the goal of winning one high value item at a low price. In this mechanism, we will use $SW(s)$ to denote the social welfare achieved by action profile $s$, the sum of player utilities plus the auctioneer's revenue, and use $\text{OPT}^t$ is the maximum social welfare possible with players in round $t$.

The first price item auction is a $(1-1/e, 1)$ smooth mechanism and hence has a price of anarchy of $e/(e-1) \approx 1.58$. Lykouris et al. [19] prove that a price of anarchy of $3.16(1+\epsilon)$ is guaranteed if players use adaptive learning and the turnover probability is at most $p \leq \epsilon^2\gamma^2/(\log(dT)\log(1/\gamma))$, which corresponds to $\rho = 2$ and $\kappa = \log(1/\gamma)$. Using the proof from [19] with the improved $A(d,T)$ term of Proposition 7, we get an $\gamma^{-1}$ improvement in the probability term.

**Theorem 18.** *If all players use Low Approximate Regret algorithms for shifting experts with parameters $\eta$ and $A(d,T) = \log(dT)$ in a dynamic population matching market with first price item auctions, then*

$$3.16(1+\eta)\sum_t \mathbb{E}[SW(s^t)] \geq \sum_t \mathbb{E}[\text{OPT}^t], \qquad (24)$$

*assuming the turnover probability $p$ has at most $p \leq \epsilon^2\gamma/(\log(dT)\log(1/\gamma))$.*

In other games and mechanisms [19] including congestion games, bandwidth-sharing, and large markets, we achieve analogous improvements.

## C.2 Proof of Proposition 7

Noisy Hedge is a modification of Hedge that mixes the distribution returned by the exponential update with a small uniform noise at each step. Fix $\theta \in [0,1]$, $\eta > 0$, and let $\pi$ be the uniform distribution over $[d]$. Let $w^1 = \pi$. Then the Noisy Hedge update at time $t$ is given by:

1. $\widetilde{w}_i^{t+1} = w_i^t e^{-\eta c_i^t}$.
2. $g_i^{t+1} = \widetilde{w}_i^{t+1}/\sum_{j\in[d]} \widetilde{w}_j^{t+1}$.
3. $w^{t+1} = (1-\theta)g^{t+1} + \theta\pi$.

**Lemma 19.** *Let $f^1, \ldots, f^T \in \Delta(d)$ be any sequence of experts with $K$ changes. Then for any sequence of costs $c^1, \ldots, c^T \in [0,1]^d$, Noisy Hedge with learning rate $\eta > 0$ and $\theta = 1/T$ enjoys the regret bound*

$$\sum_{t=1}^T \langle w^t - f^t, c^t\rangle \leq \eta\sum_{t=1}^T \langle w^t, c^t\rangle + \frac{1}{\eta}(2\log d + K\log(dT)).$$

**Proof of Lemma 19.** We follow a proof similar to that of Hedge (Proposition 12). Note that we have

$$\langle w^t - f^t, c^t\rangle = \langle w^t - \widetilde{w}^{t+1}, c^t\rangle + \langle \widetilde{w}^{t+1} - f^t, c^t\rangle. \qquad (25)$$

For the first term we may reuse the following bound from the proof of Proposition 12:

$$\langle w^t - \widetilde{w}^{t+1}, c^t\rangle \leq \eta\langle w^t, c^t\rangle. \qquad (26)$$

For the second term, as in Proposition 12, we use the inequality:

$$\langle \widetilde{w}^{t+1} - f^t, c^t \rangle = \frac{1}{\eta}\Big(D_R(f^t \mid w^t) - D_R(f^t \mid \widetilde{w}^{t+1}) - D_R(\widetilde{w}^{t+1} \mid w^t)\Big)$$

$$\leq \frac{1}{\eta}\Big(D_R(f^t|w^t) - D_R(f^t|g^{t+1})\Big),$$

where the Bregman divergence is the KL divergence, i.e. $D_R(f|g) = \sum_j f_j \log(f_j/g_j)$. Summing over all $t$, we have:

$$\sum_{t=1}^{T} \langle w^t - f^t, c^t \rangle \leq \eta \sum_{t=1}^{T} \langle w^t, c^t \rangle + \frac{1}{\eta}\sum_{t=1}^{T}\Big(D_R(f^t|w^t) - D_R(f^t|g^{t+1})\Big) \qquad (27)$$

To bound the second term, we distinguish between three cases. First, the term $D_R(f^1|w^1)$ can be bounded as in Proposition 12 by $\log(d)$ since $w^1$ is the uniform distribution.

Second, at some $t > 1$ where a change in the comparator occurred ($f^t \neq f^{t-1}$), we can bound $D_R(f^t|w^t)$ by $\log(d/\theta)$ since $w^t$ has is at least $\theta/d$ due to the mixing of the noise. This is exactly the reason why we need the noise — this term could be unbounded otherwise.

Last, for some $t > 1$ when the comparator did not change ($f^t = f^{t-1}$), we bound $D_R(f^t|w^t) - D_R(f^{t-1}|g^t)$ by $\theta \cdot d$. To prove that, note that since without loss of generality $f^t$ is an indicator vector, there is only one summand we are interested in the Bregman divergence. Let's call this summand $j$. What we want to bound is hence

$$D_R(f^t|w^t) - D_R(f^{t-1}|g^t) = \log(1/w_j^t) - \log(1/g_j^t) = \log(g_j^t/w_j^t).$$

As a result:

$$\sum_{t=1}^{T}\Big(D_R(f^t|w^t) - D_R(f^t|g^{t+1})\Big) \leq \log(d) + T\theta \log(d) + K \log(d/\theta). \qquad (28)$$

Combining inequalities 26 and 28 and setting $\theta = 1/T$, the result follows. □

## C.3 Proof of Proposition 8

The proof of Proposition 8 is analogous to that of Proposition 1.

Recall that $s^{*1:T}$ is a solution sequence with cost at most $\rho$ times the minimum cost that is relatively stable to the turnover of players and that this sequence can be randomized.

For such a sequence of solutions, we use $K_i$ to denote the sum of total variation distances $K_i = \sum_t \big\| s_i^{*t} - s_i^{*t-1} \big\|_1$ of the strategy for player $i$ in this sequence.

$$
\begin{aligned}
(1-\epsilon)\sum_{t=1}^{T} \mathbb{E}\big[C(s^t)\big] &= (1-\epsilon)\sum_{i\in[d]}\sum_{t=1}^{T} \mathbb{E}\big[\mathbf{cost}_i(s^t)\big] \\
&\leq \sum_{i\in[d]}\left[\sum_{t=1}^{T} \mathbb{E}\big[\mathbf{cost}_i(s_i^{*t}, s_{-i}^t)\big] + \frac{1+\mathbb{E}[K_i]}{\epsilon}A(d,T)\right] \\
&\leq \sum_{t=1}^{T}\big(\lambda\,\mathbb{E}\big[C(s^{*t})\big] + \mu\,\mathbb{E}\big[C(s^t)\big]\big) + \frac{n+\mathbb{E}[\sum_i K_i]}{\epsilon}A(d,T).
\end{aligned}
$$

Here we are taking expectation over the randomness in $s_i^{*1:T}$ due to players turning and/or due to randomness in the approximate minimization algorithm. The first inequality holds because each player satisfies the Low Approximate Regret property (22) for total variation distance, applied with $s_i^{*1:T}$ as the comparator sequence. As was discussed in Appendix C.1, the property (22) is implied by Low Approximate Regret for shifting experts (2). The second inequality follows from smoothness.

The claimed bound follows by rearranging terms. □

# D   Utility Maximization Games and Mechanisms

In this section, we show how all our results extend to utility maximization games and mechanisms.

Consider a static game $G$ among a set of $n$ players. Each player $i$ has an action space $S_i$ and a utility function $\textbf{utility}_i : S_1 \times \cdots \times S_n \to [0,1]$ that maps an action profile $s = (s_1, \ldots, s_n)$ to a utility $\textbf{utility}_i(s)$. The goal of each player is to maximize their utility. One can simply adapt our definitions of Low Approximate Regret by treating utilities as negative costs. While one might imagine applying the same strategy to adapt algorithms to the utility setting, extra care is required. Not all algorithms necessarily admit such a direct adaptation (or adapt at all). However, all the algorithms analyzed in this paper do, and their proofs are designed to carry through with this adaptation. We demonstrate this by sketching the proofs for Hedge and Algorithm 3 of Low Approximate Regret with utilities, but the same holds for all the other algorithms we analyze.

As in the cost minimization setting, we assume that at each round $t$, player $i$ picks a probability distribution $w_i^t$ and draws her action $s_i^t$ from this distribution. The utility she receives when playing action $x$ is $u_{i,x}^t = \textbf{utility}_i(x, s_{-i}^t)$ where $s_{-i}^t$ is the set of strategies of all but $i^{\text{th}}$ player. Let $u_i^t = (u_{i,x}^t)_{x \in S_i}$.

An important class of utility maximization games are mechanisms, such as auctions, where money plays special role. The players' actions $s_i$ typically involve bidding on items, and the outcome of an action profile $s$ comes in two parts: $v_i : S_1 \times \cdots \times S_n \to [0,1]$, which is the resulting value for player $i$, and $p_i : S_1 \times \cdots \times S_n \to [0,1]$, which is the price player $i$ has to pay. Her utility is then $\textbf{utility}_i(s) = v_i(s) - p_i(s)$.[5] We evaluate such mechanisms via the notion of social welfare $SW(s) = \sum_i v_i(s)$, the sum of the utilities of the players plus all the payments; this is the revenue of the mechanism. A simple example of such a mechanism is the first price auction: The player's strategy is a bid, and the highest bidder wins the item and pays her own bid.

We use the the smooth mechanism definition of [29].[6]

**Definition 3** (Smooth mechanism [29]). *A utility maximization mechanism is called $(\lambda, \mu)$-smooth if there exists a strategy profile $s^\star$, such that for all strategy profiles $s$: $\sum_i u_i(s_i^\star, s_{-i}) \geq \lambda\text{OPT} - \mu \sum_i p_i(s)$, where $\text{OPT} = \max_{s^o} \sum_{i=1}^n \textbf{utility}_i(s^o)$.*

Note the slight difference from Definition 2. In proving the price of anarchy property we used the game's smoothness property with $s^*$ as the action profile resulting in OPT total cost. In the definition for mechanisms, we do not insist that $SW(s^*) = \text{OPT}$.

Recall from section 2 that first price item auctions are $(1 - 1/e, 1)$-smooth and all-pay actions are $(1/2, 1)$-smooth. We show in Proposition 20 that smooth mechanisms have a price of anarchy of at most $\max(\mu, 1)/\lambda$.

**Definition 4** (Low Approximate Regret for utility maximization). *A learning algorithm for player $i$ that uses action distributions $w_i^t$ in step $t$ satisfies the Low Approximate Regret property for a parameter $\epsilon$, and a function $A(d, T)$ if for all action distributions $f \in \Delta(S_i)$:*

$$(1 + \epsilon) \sum_{t=1}^T \langle w_i^t, u_i^t \rangle \geq \sum_{t=1}^T \langle f, u_i^t \rangle - \frac{A(d, T)}{\epsilon}. \tag{29}$$

*An algorithm satisfies Low Approximate Regret for the shifting experts setting if for all sequences $f^1, \ldots, f^T \in \Delta(S_i)$, letting $K$ be the number of shifts, i.e. $K = |\{t > 2 : f^{t-1} \neq f^t\}|$:*

$$(1 + \epsilon) \sum_{t=1}^T \langle w_i^t, u_i^t \rangle \geq \sum_{t=1}^T \langle f^t, u_i^t \rangle - (1 + K)\frac{A(d, T)}{\epsilon}. \tag{30}$$

*We say that an algorithm satisfies the Strong Low Approximate Regret property if it satisfies (29) or (30) for all $\epsilon > 0$ simultaneously. In the bandit feedback case, we require the property to hold in expectations over the realized strategies of player $i$.*

Now we are ready to prove the utility maximization analog of Proposition 1

**Proposition 20** (Efficiency for Mechanisms). *Consider a $(\lambda, \mu)$-smooth mechanism. If all players use Low Approximate Regret algorithms satisfying Eq. (29) for parameter $\epsilon$, then*

$$\frac{1}{T} \sum_t \mathbb{E}\big[SW(s^t)\big] \geq \frac{\lambda}{\max(\mu, 1+\epsilon)} \text{OPT} + \frac{n}{T} \cdot \frac{1}{\max(\mu, 1+\epsilon)} \cdot \frac{A(d, T)}{\epsilon}.$$

*where $s^t$ is the action profile drawn on round $t$ from the corresponding mixed actions of the players.*

**Proof.** We get the claimed bound by considering $(1 + \epsilon) \sum_t \mathbb{E}[\sum_i \mathbf{utility}_i(s^t)]$, using the low approximate regret property with $f = s_i^\star$ for each player $i$ for the action $s^\star$ in the smoothness property, then using the smoothness property for each time $t$ to bound $\sum_i \mathbf{utility}_i(s_i^\star, s_{-i}^t)$, and rearranging terms. $\square$

**Proposition 21.** *Hedge with a constant learning rate and uniform prior over actions satisfies the utility version of the Low Approximate Regret property with $A(d, T) = (e - 1) \log(d)$.*

We mirror the proof of Proposition 12 with $c^t = -u^t$. The only place where the analysis does not automatically go through is where we need that the costs are in $[0, 1]$, namely equation (9). Note that the first inequality there ceases to hold when $c^t < 0$. However it is still the case that $1 - e^{-\eta x} \leq (e - 1)\eta x$ for $x \in [-1, 0]$. Hence we have:

$$\langle w^t - \widetilde{w}^{t+1}, c^t \rangle \leq \eta(e - 1) \sum_{j \in [d]} w_j^t (c_j^t)^2 \leq \eta(e - 1) \sum_{j \in [d]} w_j^t (-c_j^t).$$

The last inequality holds as $-c_j^t \in [0, 1]$.

With this inequality, combined with the rest of the proof in Proposition 12, we have: $\sum_t \langle w_t - f, c^t \rangle \leq \eta(e - 1) \sum_t \langle w^t, -c^t \rangle + \frac{\log(d)}{\eta}$. Setting $\epsilon = \eta(e - 1)$ and substituting $c^t$ yields

$$(1 + \epsilon) \sum_{t=1}^T \langle w^t, u^t \rangle \geq \sum_{t=1}^T \langle f, u^t \rangle - \frac{(e - 1) \log(d)}{\epsilon}.$$

which proves the claim.

**Bandit Feedback** We now provide some more discussion regarding Algorithm 3, since the improvement on the number of strategies occurs in utility maximization settings.

The algorithm's update step for utilities is obtained by using $c^t = -u^t$, but note that the normalization factor is $\gamma \geq 0$ for utility settings.

The Low Approximate Regret proof is achieved as in Lemma 6 again by replacing cost with negative utility.

**Lemma 22** (Regret bound for Algorithm 3 with utilities). *For any $f^\star \in \Delta(d)$, and any sequence of utilities $u^1, \ldots, u^T \in [0, 1]^d$, the weights generated by Algorithm 3 with $\eta \in (0, 1)$ satisfy*

$$\mathbb{E}\left[\sum_{t=1}^T \langle w^t, u^t \rangle\right] \geq \mathbb{E}\left[\sum_{t=1}^T \langle f^\star, u^t \rangle\right] - \frac{\eta}{1 - \eta} \mathbb{E}\left[\sum_{t=1}^T \langle w^t, u^t \rangle\right] - \frac{1}{\eta} d \log(T/d) - d. \quad (31)$$

**Proof of Lemma 22.** Define a cost sequence $c^1, \ldots, c^T$ via $c^t = -u^t$ and run Algorithm 3 with these costs. From Lemma 16, we have that for each $t$,

$$-\langle w^t - f^\star, \hat{u}^t \rangle \leq \eta \sum_{j \in [d]} \frac{(w_j^t \cdot \hat{u}_j^t)^2}{1 + \eta w_j^t \hat{u}_j^t} + \frac{1}{\eta}\big(D_R(f^\star \mid w^t) - D_R(f \mid w^{t+1})\big),$$

where $\hat{u}_j^t = \mathbb{1}\{j = s^t\} u_j^t / w_j^t$. Applying an analysis identical to that of Lemma 17 on this bound yields the result. $\square$

## Footnotes

[5]We assume that all $s$ have $v_i(s) - p_i(s) \geq 0$.

[6]For the dynamic population game setting we use a variant of this definition, *solution-based smoothness*, where OPT in the RHS is replaced by the social welfare of a near-optimal solution as in [19].