[Reviews · NeurIPS 2016]

Reviewer 1

Summary

The paper presents an analysis of a wide class of learning algorithms in smooth games, demonstrating fast convergence. This is an interesting result within game-theoretical control - traditional results prove convergence, but the rate is somewhat new. The authors have brought together techniques from learning theory, with results based on price of anarchy in smooth games. The results are interesting to the game-theoretical control community.

Qualitative Assessment

I enjoyed the paper, as somebody who knows both game theory and learning theory. The results are novel and interesting, and I think important for the area of game-theoretical control. There is a lot of material here - fast learning for both full information feedback and bandit feedback, in expectation and in high probability, and in addition similar results in dynamic population games. However this is, in essence, the major failing of the paper. So much has been crammed in to such a short space that the work is poorly motivated, terms are not well defined, and I struggle to reconcile the specialist knowledge needed to appreciate the work with a general conference such as NIPS. Post-rebuttal: Given my complaints of too much material, and your (fair) rebuttal, perhaps you could try to amend the intro slightly to emphasise more that you feel your major contribution is demonstrating the benefit of considering low approximate regret, and to provide examples of the power it gives - I (incorrectly it turns out) read the paper as if you were most pleased about being able to show Fast Convergence of Common Learning Algorithms in Games. (Perhaps a title change should be considered?)

Confidence in this Review

3-Expert (read the paper in detail, know the area, quite certain of my opinion)


Reviewer 2

Summary

The paper presents a number of new results in the decentralized no-regret dynamics on smooth games. The authors propose a new type of property called "Low Approximate Regret", generalizing previous classical algorithms. New results also include different feedback machanisms, the bandit case, and population games.

Qualitative Assessment

This paper presents a number of new results that follow-up on SALS15 (and LST16). The results are significant and interesting to this community. The main contributions (improved convergence bounds and in the different feedback settings) are possible due to a reformulation of no-regret called The Low Approximate Regret property, which bounds the difference between the approximate (1-epsilon) cost paid and the comparator. The authors show that this property holds for a number of well-known learning algorithms and shows regret bounds based on this property. If there is any problem with the paper, it's that it contains too much content. As a result, the bulk of the derivations are in the appendix and only the main points are summarized in the paper. This makes the paper a nice read and results easier to contextualize with related work in the area. However, it does feel that there are frequent jumps to different settings and topics throughout the paper. Also, though it is not a requirement, it is always nice to have experiments to complement/validate theoretical results and this paper has none, whereas SALS15 also showed the practical application. The paper ends also ends abruptly with a conclusion. One point of clarification: the authors use the term "realized feedback" and then later use the term "full information feedback" to mean the same. Consistent terminology should be used throughout the paper; if these are not the same, please explain the difference. In the bandit case, the authors should distinguish between the 'expected Low Approximate Regret' in Lemma 4 and Low Approximate Regret in Equation (1).

Confidence in this Review

2-Confident (read it all; understood it all reasonably well)


Reviewer 3

Summary

This paper introduces a property called low approximate regret for the expert algorithm that will ensure faster convergence in games compared to previous work. The authors show that this property can be satisfied by many common algorithms, including the vanilla Hedge algorithm. The feedback model allowed in this paper is also more natural and less restricted: the player only receives a realized loss vector depending on other players’ chosen actions, or even just the chosen coordinate of this loss vector (i.e. bandit feedback). High probability bounds are also derived. Finally the authors also extend the results to dynamic population games and improve upon previous work by [LST16].

Qualitative Assessment

The results in this paper are interesting in general. However, from a technical viewpoint, the main results seem to be a simple generalization of what is discussed in Thm 23 of [SALS15] (on how a small loss bound leads to a faster rate). Indeed, small loss bound implies low approximation regret as discussed in Sec 3 of this paper. Furthermore, although the main results in [SALS15] are about the expectation feedback model, their result in Thm 23 is actually applicable to the same realized feedback model. So the claim that this paper improves upon [SALS15] in terms of the feedback model seems to be an overstatement to me. In fact, the claim about the improved speed is also a bit misleading, at least for the earlier part of the paper. It does convergence faster, but only to an approximation of the PoA. So in some sense these convergence rates not exactly comparable. In the bandit setting, there is actually a known algorithm that gives a small loss bound against non-oblivious adversary. See Allenberg, Chamy, Peter Auer, László Györfi, and György Ottucsák. "Hannan consistency in on-line learning in case of unbounded losses under partial monitoring." 2006. It looks like this can be directly applied here? Some detailed comments are listed below: 1. I would suggest explaining why most of the previous work requires the expectation feedback model while it’s not necessary here. 2. As mentioned above, Line 49 (the second bullet point for improvements) seems to be an overstatement. Same for Line 80, “without a constant factor loss” but still not exactly the PoA, right? 3. In Definition 1, maybe \epsilon should be explicitly required to fall in (0,1)? 4. In Definition 2, the right hand side of the inequality is missing the sum over i. 5. In the end of Proposition 2, instead of writing \epsilon in terms of \gamma, maybe it’s more natural to write \gamma in terms of \epsilon? 6. I would suggest putting Example 1-3 (and related paragraphs) to Sec 2, right after giving the definition of low approximate regret. 7. I don’t see how doubling trick can make any algorithm with weak low approximate regret to an algorithm with strong low approximate regret. 8. For Theorem 3, it’s weird to use informal statements like “regret is bounded by \sqrt{Total loss of best action}” in a theorem. I suggest just writing down the exact expression. 9. Line 222, why “within a (1+\epsilon) favor”? It’s \lambda/(1-\mu-\epsilon) compared to \lambda/(1-\mu), right? 10. The proposed noisy Hedge is a bit similar to Fixed Share, see for example Cesa-Bianchi, et al. 2012 "Mirror descent meets fixed share (and feels no regret)." 11. Some typos and small mistakes: 1) Line 3, “a small a multiplicative” 2) Line 34, remove “in”? 3) Line 42, “…more information than is…”? 4) Line 155, “required that is s is…”? 5) Line 207, “…For instance [FS97, YEYS04]…”, maybe should put this into a parenthesis, and make the punctuation right.

Confidence in this Review

3-Expert (read the paper in detail, know the area, quite certain of my opinion)


Reviewer 4

Summary

The authors study repeated games where players use regret minimizing algorithms for choosing their actions, generalizing some recent work in significant ways. In particular, they require only "realized feedback" as opposed to the expectation over actions of other players -- a much more realistic setting. A few nice extensions and observations about bandit feedback and dynamic population games are also presented.

Qualitative Assessment

This was a very nice paper that was well written, the problem clearly motivated, and improvements on prior work (SAL15 and LST16) are conceptually significant, despite the fact that the technical aspects seem rather similar. In particular, the requirement of "realized feedback" instead of "expected feedback" is significant as it brings the problem into a much more realistic setting. I would have liked to see some experimental comparison with SAL15 / LST16 -- it is not yet clear to me if lacking this assumption has an effect in practice. In particular, if I *do* have access to expected feedback, should I use it? ** Thanks for your response to this in the rebuttal. Please do add your empirical results in the final version of the paper. ** Overall a nice piece of work which would be of interest to game theorists.

Confidence in this Review

2-Confident (read it all; understood it all reasonably well)


Reviewer 5

Summary

The paper focus on a relaxation of the standard no-regret property in online algorithms which they define as a low approximate regret property and which allows a small a multiplicative approximation factor of regret. This property is satisfied by many learning algorithms and the flexibility of the multiplicative term in the definition allows for a smaller error term, since the standard error term can be "rolled" in the multiplicative factor. This approximate low regret outcomes still provide price of anarchy guarantees in (\lamba-\mu) smooth games and the small error term after a smaller number of period of play. The results require only realized feedback, or even just bandit feedback in some cases, not the expectation over actions of other players,

Qualitative Assessment

I think the trick of additive error term to the multiplicative error term is interesting and is exploited well in the paper. There is a lot of technical overlap with both SAL15, LST16 which makes me less enthusiastic about an oral level presentation. Also, I think the paper language to some extent is overselling the result to the point where it becomes a distraction. Does learning really converge fast (to equilibria) in games? The main point here is that if we relax the notion of convergence (from convergence of the actual behavior to convergence of time-averages) and if we relax the notion of target states (to be states where agents have low approximate regret) the (\lambda-\mu) smoothness results still carry over. The technical points are interesting in themselves. Stretching the interpretation of convergence, equilibrium to become increasingly inclusive so as to get faster "convergence" detracts from these points. Also, if this is a result for general games (as the title suggests) what are the practical implications for the class of two player games? Or even for 2x2 games?

Confidence in this Review

3-Expert (read the paper in detail, know the area, quite certain of my opinion)